# A Study on Two Conditions for the Realization of Artificial Empathy and Its Cognitive Foundation

Zhongliang Cui [1,2,*] and Jing Liu [3]

1 Research Center for Artificial Intelligence Epistemology, Nanjing University of Information Science & Technology, Nanjing 210044, China
2 Department of Philosophy, East China Normal University, Shanghai 200062, China
3 School of Foreign Languages, Zhoukou Normal University, Zhoukou 466001, China
* Correspondence: cuizhongliang1986@126.com

**Abstract:** The realization of artificial empathy is conditional on the following: on the one hand, human emotions can be recognized by AI and, on the other hand, the emotions presented by artificial intelligence are consistent with human emotions. Faced with these two conditions, what we explored is how to identify emotions, and how to prove that AI has the ability to reflect on emotional consciousness in the process of cognitive processing, In order to explain the first question, this paper argues that emotion identification mainly includes the following three processes: emotional perception, emotional cognition and emotional reflection. It proposes that emotional display mainly includes the following three dimensions: basic emotions, secondary emotions and abstract emotions. On this basis, the paper proposes that the realization of artificial empathy needs to meet the following three cognitive processing capabilities: the integral processing ability of external emotions, the integral processing ability of proprioceptive emotions and the processing ability of integrating internal and external emotions. We are open to whether the second difficulty can be addressed. In order to gain the reflective ability of emotional consciousness for AI, the paper proposes that artificial intelligence should include consistency on identification of external emotions and emotional expression, processing of ontological emotions and external emotions, integration of internal and external emotions and generation of proprioceptive emotions.

**Keywords:** artificial empathy; cognitive processing; external emotion; proprioceptive emotion; emotional reflection

## 1. Introduction

The key feature of human interaction is to infer the emotional state of others based on their covert and/or overt signals [1] and Heidegger regarded human emotion as the authentic state of human existence. Artificial intelligence draws on empathy research and proposes the concept of artificial empathy, arguing that human fear of artificial intelligence stems from imagining the emergence of a society lacking empathy, and having empathy has become one of the most important issues in artificial emotion research. However, the biggest question about artificial emotions is whether artificial intelligence can fully simulate and recognize human emotions. Even if this can be simulated, it still faces the question of whether it has an empirical basis for emotion recognition. Affective computing research represented by R.W. Picard attempts to solve these problems: "emotional computing captures emotions through physiological data such as facial expressions, body expressions, language, text, skin conductance and heart rate, as well as senses such as touch" [2]. However, affective computing mainly focuses on the emotions expressed by individuals at the physiological level [1], it is more difficult to calculate the group level and psychological level, and the emotional interaction between humans and machines is even more difficult to understand. Based on the embodied development of philosophy and cognitive science, at the same time, in order to promote further research into this problem, this study tries to

remove the shackles of individualism and cognitivism in the research on artificial emotion. Individualism makes the study of artificial emotion subject to the individualization of emotion, so that the expression of artificial emotion lies more in the imitation of human emotion than the interaction with human beings. Cognitivism places the understanding of artificial emotions more in the representation and calculation of human emotions, so it also lacks attention to the generation of emotions in human interaction. In a word, individualism and cognitivism cause the study of artificial emotion to fall into a complex process of cognitive processing and lack the study of direct perception of emotion. In addition, this paper attempts to provide a solution from the level of artificial empathy and directly enter the interaction level of human–computer emotion through the conversion of sight. It is necessary to clarify several different concepts of emotional interaction here. The roles of sympathy, compassion and empathy in effective support is well known. The reason why sympathy or compassion was not chosen as the basis for an artificial emotional realization is because "sympathy and compassion are getting support from friends, family and mental health services, whereas empathy arises in relationships with others who are suffering or have similar pains and life difficulties" [3]. Empathy emphasizes interaction and relationship. Therefore, on the one hand, a machine can quickly become a part of human life through the practice of artificial empathy and, on the other hand, it can also become a support of daily emotions.

## 2. Two Basic Conditions for the Realization of Artificial Empathy

With the development of artificial intelligence in recent years, research on affective computing is expanding rapidly, and research on artificial intelligence based on emotion has gained rapid development. "By 2024, the affective computing market is expected to reach 90 billion dollars" [3]. At the same time, the development of phenomenology and embodied cognitive science also shows that human nature does not only consist of independent self-centeredness, as has long been assumed in traditional Western psychology. The premise of self-formation includes a moral dimension of empathy [4] and the individual's cognition of emotion is achieved through the ability to simulate emotion or partial re-experience of emotion [5]. Therefore, artificial empathy cannot be realized by the current individual-centered affective computing. On the one hand, it faces the difficulty of emotion recognition and, on the other hand, it needs to realize the emotional consistency between human and machine. Artificial empathy can be divided into the following three categories:

(1)　Action empathy, an automatic non-cognitive process that manifests as the convergence of machines to human emotions in expressions, voices, and gestures;
(2)　Emotional empathy, the emotional response of machines oriented by human emotional states;
(3)　Cognitive empathy, the machine's ability to identify human emotional states, enable transpositional thinking and reasoning. However, this study pays more attention to the emotional state that emerges in the process of human–computer interaction as a whole, and therefore does not divide empathy into different levels.

### 2.1. Human Emotions Can Be Recognized by Artificial Intelligence

In order to better integrate artificial intelligence into human emotional life, artificial emotion research is introducing new quantitative emotion knowledge systems and corresponding emotion processing models. However, if artificial emotion research is still limited to individualism to realize emotion recognition, it will face two unsolvable problems: on the one hand, the variability of emotions leads to the undecidability of emotional intentions and, on the other hand, emotion recognition is the same as the problem of other minds, which cannot really achieve full access to other minds. Phenomenology provides a good explanation for the problem of emotion recognition from the perspective of empathy. Stein divides empathy into the following three processes: direct perception, empirical projection, and mentalized interpretation. Although these types of empathy are all knowledge of external experience, only the latter (mentalization) can properly be called a kind of intel-

lectual knowledge [6]. Therefore, the realization of empathy is based on direct perception, while the projection and mentalization of experience are complementary to the realization of empathy, and emotion is the emergence of social interaction. Emotional changes and interpersonal interactions form the same frequency resonance, eliminating the problem of emotional variability. Empathy is the immediate experience of another person's experience, and another part of the same act, while knowing that it is another person's experience [6], a feeling that one person is aware of another person, and then vice versa [7]. Therefore, it can be seen that empathy pulls emotion from the individual into the intersubjective state, which eliminates the problem of other minds faced by emotion recognition, and also advances the recognition and calculation of emotion to the stage of emotion perception and expression. Then, whether human emotion can be recognized by artificial intelligence requires the help of emotion perception ability, which requires the conversion of the traditional analysis-based cognitive processing model of emotion to the whole-based perception processing model of emotion.

For the perception of emotion, multi-modal synaesthesia is an advantageous option. Perception is the core of the internal relationship between all objects and entities. The recognition of human emotions by artificial intelligence requires a perception-based emotion recognition method. Therefore, perceptual ability is the basis of human life and is also the realization of human–computer interaction. Breaking away from the traditional concept of strictly distinguishing perception and cognition, embodied cognitive science believes that perception contains understanding, and perception itself is understanding. The two are not separate processes, but the subsequent artificial binary separation that leads to understanding as an independent internal mechanism. For empathy, Stein clearly stated that empathy is a perceptual behavior, and people directly perceive and understand each other's emotions as a way of perception, but not through reasoning. Emotional experiences are very complex, including one's beliefs about the world, oneself and others, as well as many preferred scenarios and outcomes. When we have emotions, it is obvious how and why we may not be able to identify them [8]. To illustrate the understanding of this emotion-led perception bias, Picard cites a range of perception-related theories, in particular Cytowic's research on synaesthesia [3]. That is to say, artificial intelligence can use synaesthesia as an important means of perception to enhance the ability to perceive emotions; at least when acquiring physiologically relevant emotional information, synaesthesia is an important type of perception. Synaesthesia perception patterns and multisensory integration of affective information have important implications for affective recognition. Integrating auditory and visual, nonverbal affective information contributes to affective recognition. The recognition of audiovisual stimuli has higher accuracy and faster reaction times than the recognition of visual or auditory stimuli alone [9]. Therefore, multi-modal synaesthesia perception will accelerate the recognition of emotion by artifitcial intelligence and further improve the accuracy of emotion recognition.

The content of emotion perception includes the following three aspects: external emotional information perception, processing-centered emotional experience and ontology emotional information perception. Ontology perception and the processing center work together to support the perception of external emotions. In the process of emotion perception, the embodied theory of emotion further analyzes the emotional content in detail, and believes that emotion perception has the following three steps [10]:

(1) The perceiver subtly imitates the facial expression of the target person;
(2) Subtle muscle contractions of the perceiver's face generate afferent muscle feedback from the face to the brain;
(3) The perceiver uses this feedback to reproduce the emotion, thereby understanding the meaning of the perceived emotion. Thus, perception is not a passive process of acquiring information about external objects, but an active processing process, through which we directly contact the world and live in it [11].

Therefore, emotions are not just displayed on other people's faces, but the entire body postures and movements carry all external emotions, including all emotional information

clues on limbs and faces. Maibom also regarded empathy as a kind of emotional matching to others' emotional intentions [12]. The matching indicates that emotional content will realize emotional coupling between the self and others on the basis of perception, rather than analysis and representation based on cognitive reasoning.

### 2.2. The Emotions Presented by Artificial Intelligence Are Consistent with Human Emotions

A prerequisite for artificial intelligence to achieve empathy is that it can show emotional expressions similar to human emotions. For a completely different emotional experience, human beings cannot use the traditional categories to divide, so it is then impossible to realize the emotional understanding of the objective, and it is impossible to easily integrate this object into human life and realize empathy. Empathy is an emotional response that is similar to how others feel or are expected to feel in a particular situation [7]. Of course, with the current technology is still very difficult for artificial intelligence to achieve exactly the same human emotions. In order to solve this paradox, we need to explore the type of emotional interaction in the process of empathy. Therefore, it is a prerequisite for artificial intelligence to have emotional ability to be accepted by humans and enter the social system, but learning to make artificial intelligence also show similar emotional expression to humans is the basis for artificial intelligence to achieve empathy. The emotions of artificial intelligence are similar to human emotions. Artificial intelligence needs to understand the relevant emotional state of human beings, and the emergence of this state requires the emotional experience of continuous communication and proprioception with human beings, resulting in human-like emotions and the ability to respond in appropriate ways during interactions. The artificial intelligence agent Elefriends, by connecting a counselor with other mentally ill patients with the same experience, enables them to experience a kind of peer support, so as to realize the occurrence of empathy [3]. Therefore, a consistent emotional state and emotional experience will promote empathy between humans and machines.

However, it is also important to note that emotions are not all universally consistent, but rather diverse. Chinese subjects living in France tended to imitate both Chinese and French angry facial expressions, while French participants only tended to imitate French but not easily imitate Chinese expressions [5]. This shows that emotion is an acquired construction and needs the empirical basis of emotional life. Emotions are inseparable from beliefs, memories, perceptions, attitudes, etc. Our entire psychological life is full of emotions, which are intrinsically related to individual and collective socio-historical backgrounds [3]. The big question here, then, is how can the emotions presented by artificial intelligence be aligned with human emotions, and does that alignment need to be exactly the same? From the perspective of emotional embodiment, we can see that emotions have basic emotional manifestations in everyone, but the way they are presented is not exactly the same. Therefore, the consistency here is more of a convergence or a coupling of emotional relationship, that is, human and artificial intelligence can achieve basic emotional interaction, and do not feel artificial. Compared to genuine smiles, fake smiles tend to be more asymmetrical, have shorter onset and disappearance durations, and exhibit more irregularities such as pauses and changes in intensity [13]. Therefore, artificial emotion design needs to eliminate the falseness of emotion, so as to ensure the unity of the relationship between humans and machines in the process of empathy.

The following example illustrates the feeling of empathy between humans and machines (Elefriends) [3]:

> I: Yeah. So, I mean on a day-to-day basis, has it made you feel more supported?

> Sue: Yes, it has, yes, because I feel the people who post on there really understand how I feel and I feel I can empathize with them. Whereas if you haven´t got mental health issues, I don't know if people really realize what it´s like.

From the above example, it can be clearly seen that emotional consistency is the basis for realizing empathy. To achieve emotional consistency, artificial intelligence needs to have the ability to "feel with". There are the following three characteristics of "feeling with":

(1) Thee emotional nature of self and others is different;
(2) Two-way causal relationship;
(3) Everyone has a sense of partial ownership of emotions [7]. That is to say, both self and others have emotions, and the emotions of both parties are formed in the interaction, thus the consistency of this emotion does not mean that artificial intelligence and human performance are exactly the same in the interaction process. When we expose a smiling face to artificial intelligence, we don't expect it to respond to the same smiling face, but to be able to respond to the smile with the appropriate emotional expression and make the communication continue or break. Each emotion has its own unique characteristics: signals, physiological and antecedent events. Each emotion also has the same characteristics as the others: rapid onset, short duration, unsolicited, automatic evaluation and consistency between responses [14]. Therefore, emotional consistency is manifested as emotional sharing, that is, the meaning of interaction expressed through face, voice, gesture, posture and touch, so as to realize emotional sharing and obtain a similar emotional experience. Emotional experience arises through interoceptive sensations and physiological behaviors mediated through neural pathways [15]. In his Critique of Judgment, Kant praised the importance of shared ("intersubjective") feeling in the process of appreciating beauty [8].

In short, the realization of artificial empathy needs to eliminate the traditional analytical interpretation of individualism and rationalism, enter the field of embodied emotion between subjects, and return empathy to the dynamic process of human–computer interaction. Such emotional interaction, on the one hand, needs to satisfy the recognition of emotion by artificial intelligence relying on perception, and realize the complete possession of emotion through multi-modal synaesthesia perception of the whole emotion. On the other hand, the emotional integration and multicultural adaptation between humans and machines can be realized based on the emergence of inter-subject interaction. Therefore, it is necessary to deeply explore the way of expression of emotion, based on the perception of interaction between subjects, so as to ensure the consistency of artificial intelligence in the recognition and expression of emotion.

## 3. Three Categories of Affective Display

Emotion is the inner relationship between humans, the world and society. Scientific research on emotion was first started by William James. The affective concept is not only the basis for understanding the social world, but also the basis for the development of individual behavior [16]. In fact, the study of emotion began in the time of Socrates, but emotion is mainly discussed as an opposite of rationality. Aristotle, in Rhetoric, defined emotion as "causing a great change in a person's condition, affecting judgment and accompanying pleasure and pain", Hume defined emotion as a specific feeling or "impression" [8]. By examining the progress of scientific research on emotion, it appeears that emotions can be further divided into basic categories, secondary categories and abstract categories on the basis of traditional basic emotions and complex emotions (social emotions) [2]; however, emotions are not separate emotional states, but a relationship of family resemblance.

### 3.1. Basic Emotions

The study of basic emotions began in Darwin's The Expression of the Emotions in Man and Animals in 1872, and was later absorbed and further developed by Paul Ekman, who argued that basic emotions are fundamentally differentiated, as well as innate and combinatorial [2]. Basic emotions have obvious category boundaries, including anger, disgust, fear, sadness and happiness, etc. Therefore, basic emotions are not only different emotional manifestations of individuals, but also the basis for individuals to develop complex emotions. The proposition of basic emotions is not to deny the social

constructability of emotions, but to show that basic emotions have generality from the perspective of social evolution. The social construction theory of emotion emphasizes the history of individuals, while the social evolution theory of emotion focuses on the history of species [2]. For basic emotions, Descartes favored a value-oriented analysis of emotions, believing that emotions are part of passion, and distinguishing six "primitive" passions—surprise, love, hate, desire, joy and sadness [8]. Descartes supports the basic emotional constitution, but the problem with Descartes is that all these emotions are based on the rational construction of cognition, so other complex emotions are constructed by rationality as a binder, denying the evolution and precipitation of emotions.

Basic emotions have a process of experiencing precipitation. On the one hand, this can ensure that basic emotions have meaning in life and, on the other hand, the purpose of emotions is to adapt to human survival. When we see an object that evokes childhood memories, our smile or pain is not an isolated event in the moment, but an emotional event connected by a series of experiences and meanings. This involves a temporal dimension that makes present emotions indelibly linked to past life experiences [3]. Each emotion family has the same theme, and each emotion has diversity. The former comes from natural evolution, and the latter comes from acquired reflective learning [2]. Therefore, basic emotions are the product of natural evolution and have the universal characteristics of human beings, but basic emotions are also the result of social construction and the meaningful consequence of individual development in life practice. Emotional communication between humans and machines should be based on similar basic emotions, and at the same time have acquired emotional meanings. Therefore, "secondary emotions" are also produced.

*3.2. Secondary Emotions*

In the 1880s, American psychologist William James and Danish physiologist Carl G. Lange conducted meticulous research on secondary components of emotions, such as somatic arousal, and drew the conclusion that emotions arise from the perception of the physiological state [1]. Antonio R. Damasio divided emotions into "Primary" and "Secondary" emotions. Primary emotions are equivalent to basic emotions. "Secondary" emotions are more subtle and complex. Secondary emotions are the mixture of basic emotions, which requires the participation of cognitive processes to produce [1]. Moreover, from neuroscience research on emotions, it is found that emotion processing has two parallel paths: one is faster, and the other is slower and regulated faster [17]. That is to say, the neural operating mechanism of secondary emotions and basic emotions is not consistent. Secondary emotions arise from the cognitive processing of basic emotions by the body and affect basic emotions in the opposite direction. Secondary emotions have also been confirmed in cross-cultural studies. In Chinese, the concept of emotions is more closely related to the feeling produced by afferent representations of interoceptive states; however, in English, emotional language expressions of body and behavior, including physiological responses elicited by efferent autonomic pathways, are prominent, and by extension, different systems of affective concept in language can form different cultural attitudes and emotional narratives [15]. That is to say, social and cultural differences will make individuals´ understanding and expression of emotions significantly different, and the differences will be reflected in psychological and neural processing.

From an onto-genetic point of view, new cognitive abilities emerge in the second half of the second year of life, and the emergence of consciousness or objective self-awareness (self-referential behavior) produces a new type of emotion known as "self-conscious emotion", including embarrassment, empathy and jealousy [18]. Therefore, the generation of secondary emotions is strongly related to cognitive ability. Emotion does not exist in the brain and body waiting to be triggered, but an active process based on goals [3]. Emotional expression does not depend on physical dimensions and sense of space, but on a more fluid and malleable spatial understanding of relationships. When people interact with friends on the other side of the world via social media, they feel closer than with a stranger on the bus [13]. Therefore, secondary emotions depend on perception-based higher-order

cognitive processing, and changes due to changes in mental state. When a person experiences a specific object, specific morphological states represented by perception, action, and introspection can also represent that object later offline [13]. Therefore, secondary emotion manifests as a late social construct, which is more directly related to the presence of others.

### 3.3. Abstract Emotions

The concept of abstract emotions is also a comprehensive intentional representation and cultural construction of primary emotions and secondary emotions. Activation of the body's perceptual-motor and affective systems can activate the affective concept and the emotional experience associated with the concept. There are three types of affective concepts, as listed below:

(1) The affective concepts contain information on situational antecedents or affective triggers;
(2) These concepts contain information on actions that may be taken when a particular emotion is experienced;
(3) The concepts contain information on introspection status [16]. From the perspective of developmental psychology, young children usually gradually acquire the category labels of emotions from the general conceptual level of emotions. These abstract affective concepts will become the basis for individual emotional understanding and expression, and then form a conceptual system of emotions. The division and formation of affective concepts becomes part of children's affective classification system in a very systematic order; on the contrary, the embodied state of affect indicates that the embodied states represent the core conceptual content of affect. With the emergence of emotional self-consciousness, abstract concepts related to emotions will also appear, and the generation of such concepts is not separated from the other two emotions, but generated based on basic emotions and secondary emotions. Abstract emotions are a conceptual category system formed on the basis of self-consciousness experience of basic emotions and secondary emotions.

From the perspective of embodied emotions, the generation of abstract emotions come from the image schema of emotional expression and emotional experience, which is an internal simulation of self and others´ emotions. Simulations contain the conceptual basis for knowledge of more abstract categories such as happiness and anger [19] and, during embodied simulations, emotional image is combined with bodily schemas to form emotional image schemas. Therefore, abstract emotions are the representation and synthesis of emotions associated with basic emotions and secondary emotions. "Activating conceptual knowledge about emotion can be accompanied by re-experiencing bodily states, and the simulation of sensory, motor and introspective experience forms the basis of emotional conceptual representation" [19]. Thus, abstract emotions can also be divided into more detailed categories. Niedenthal divides the affective concepts into the following three categories according to the three-level logic of categories [16]:

(1) Abstract emotion category: includes two categories of emotions, "negative" and "positive";
(2) Basic emotion category: the subordinate level seems to contain five or six basic categories, such as "love", "happiness", "anger", "sadness" and "fear";
(3) Subordinate category: reflects the finer level of the basic categories, for example, "fear" can be further subdivided into "fear/panic" and "nervous/fear". Therefore, abstract emotions are the representation and comprehensive processing of self-emotion and other people's emotion, as well as emotional expression in the process of interaction, which is the affective contents formed, together with the affective concepts.

In a word, the three-level division method of emotions represents the combination of social evolution of emotions into groups and social construction of individuals. Therefore, emotions not only have unity at the basic level, but also have diversity at the secondary level, while basic emotions and secondary emotions are unified in the emotional image schema formed in the process of individual life practice. Thus, emotions act as an intermediary to guide social practice, which makes the emotional communication of human

society smoother and closer, and also provides a foundation for the integration of artificial intelligence into human society.

## 4. The Cognitive Processing Basis of Artificial Empathy

Techniques for developing empathy and a range of emotional realizations are the holy grail of AI research [3]. The first generation of cognitive science, represented by representationalism, believes that perception and expression are isolated, resulting in an incomplete match between the theoretical driving mode of emotion and the actual expression mode. Therefore, the database scale of the data-driven mode could not cope with diverse emotional interaction environments. In contrast, embodied emotion research has shown that emotions involve different aspects of the entire body, interacting with the physical and social environment, and are not just located in the brain or nervous system [17]; thus, the idea of embodied emotion can be absorbed to eliminate the traditional artificial emotional design. Kohut defines the concept of empathy as a cognitive model based on observation and introspection, which is particularly adapted to the perception of another person's complex psychological configuration [20].

### 4.1. The Integration of Emotion Recognition and Expression

The traditional view of emotion believes that emotion is an intentional emotion expression based on the recognition and understanding of external expression. However, scientific research on embodied cognition shows that perception and expression are not two distinct processes, but two aspects of the same process. Perception resides not only in brain activity, but also in more complex dynamic coupling involving whole body activity [21]. Perception does not only occur in a special organ connected to a certain area of the brain, but is an expressive activity that includes all aspects of bodily behavior [21]. Therefore, the concept of perception–expression integration shows that the concept of emotional understanding of traditional artificial intelligence needs to be updated. Perception is not based on cognition, but on expression. Understanding without expression is incomplete. That is to say, emotional understanding relies on emotional experience, and experience is the basis of understanding. Artificial empathy under embodied emotion does not treat the emotion perception system as a separate system, because the sensory-motor system can reconstruct emotion by simulating the relevant physical state.

Emotional expression refers to the observable changes in face, voice, body and activity, and there is ambiguity between expression and state [18]. The reason for this ambiguity is the unity and indistinguishability of perception and expression. Children's ability to understand and predict their own emotions may influence their decisions about what actions to take and, in turn, this choice of action will cause or prevent certain emotional consequences [22]. Infants are also thought to behave in a way that matches emotional representations presented to them, which reveals their recognition of such representations. Infants (aged 10 weeks) laugh more and show more interest when seeing positive emotional representations and hearing pleasant sounds; however, they are more agitated and distressed when seeing adults frowning or crying [23]. Therefore, infants are able to perceive and express emotions directly, and the phenomenon of emotional contagion is a manifestation of the unity of perception and expression. Most people obtain little feedback from others about what our body movements are revealing, so we do not realize the need to monitor those movements. People usually do not adjust their bodies when they lie, because the body would be a good source of deception cues [24]. Therefore, whether it is a real emotional expression or a false emotional expression, it will be presented and perceived in human body movements.

The integration of emotion perception and expression includes the following two aspects: firstly, from the perspective of ontogeny, it is the infant's acquisition of emotion; secondly, from the research of neuroscience, the neural mechanism of the activation of perception and expression is consistent. Due to the shared empirical and neural basis of human emotion recognition and emotion expression, understanding and developing automated

systems for emotion recognition can help generate convincing human faces and/or voices with human emotion characteristics. This, in turn, produces an autonomously interacting system or agent that senses emotion and responds accordingly. The method of perception is multi-modal, because otherwise emotion or movement will be two-dimensional instead of three-dimensional, and this will make human–computer interaction closer to human interaction [1]. The discovery of mirror neurons also suggests the unity of perception and expression, with babies being able to express corresponding emotions as they look. This unity requires artificial intelligence to recognize emotion based on expression, rather than express emotion after recognition. Based on this, the emotional perception and emotional expression of artificial intelligence should be integrated. On the one hand, it is based on the same emotional model and, on the other hand, the equipment for perception and expression is integrated.

*4.2. Overall Processing Capability of External Information*

People can achieve resonance by automatically imitating other people's actions, expressions, emotions, etc. Developers believe that artificial intelligence agents can provide psychotherapy by establishing a supportive relationship with users. Chat-bots provide new opportunities to explore the emotional relationship between the body and technology, and the emergence of GPT-3 provides a realistic basis for the verbal empathy interaction between humans and machines. The value of chat-bots in psychotherapy is based on the idea of helping individuals manage and cope with their ongoing distress by promoting positive emotional connection through human–computer interaction [3]. In fact, the attribution of emotions for children is not simply triggered by recognizing living, expressive states, because they easily attribute emotions to dolls, stuffed animals, and fictional characters. All in all, children begin to express their own and others´ emotional feelings, and project such feelings onto non-human beings almost as soon as they learn to speak [22]. Therefore, children's emotions first occur on the basis of interaction with the external world and others and form emotional judgment and evaluation of things through continuous interaction.

Emotional understanding includes the concepts of valence and arousal. These two dimensions can be grouped into the following four categories: happiness/high arousal, happiness/low arousal, unhappiness/high arousal, and unhappiness/low arousal. Although it is simple, the theory allows children to place the emotions of others in these broad categories, thereby predicting the emotional quality of their subsequent actions, gaining knowledge of the positive and negative qualities of current events (such as visual cliffs), and gaining knowledge about the desires of others. This theory dominates children's thinking about emotions for most of the second and third years of life [23]. It attempts to develop machine learning-based forms of "artificial emotion", relying heavily on the assurance of authenticity through machine-readable physiological activity. Technology is empowered to bypass "unreal" emotional awareness by providing direct recognition of physical emotions [17]. Here, then, is the most fundamental question, that is, whether our physical emotions are the most authentic emotions. From the perspective of the genetics of emotions, emotions are not only physical, but also socially constructed, which requires multidimensional emotional processing. Imagine an emotionally responsive car that can alert the driver when it detects signs of stress or anger, which could impair the driver's ability to drive [1].

*4.3. Overall Processing Capability of Ontology Information*

Ontology information processing capability corresponds to external information processing and depends on the individual's deep processing of self-emotional expression and perception. Ontological information includes signals and representations of changes in physiological states, which are transformed into subjective emotional experiences and sensory states of consciousness in the brain [15]. Emotional assessments of self-awareness include pride, shame, guilt and so on. These emotions require self-awareness and the ability to compare one's behavior to standards; for example, we feel happy when winning the

lottery. However, we do not feel proud, because we do not think winning the lottery has anything to do with our actions, and the same goes for failure. If we cannot do something, we may feel bad, but if it is not our fault, then we are not ashamed or guilty [18]. Therefore, the realization of emotion processing by artificial intelligence requires the capability to experience and process emotion ontology.

Ontological information processing is also the processing of personal inner feelings. Interoceptive information refers to neural signals and representations (both non-conscious and conscious) related to the internal state of the body, including pain, temperature, bloating, itching, hunger, thirst, muscle burning, joint pain, sensory touch, flushing, visceral urgency and nausea. These feelings are derived from receptor activation in the body's visceral tissues, including pain receptors, temperature receptors, osmotic receptors and metabolic receptors [15]. The synthesis of experiences consisting of interoceptive information forms a complete autobiography of the self. Organisms can not only feel things and but also have the feeling that they can feel things. Autobiographical memories are stored in different sensory cortices and are activated by converging regions of the temporal and frontal higher cortices, as well as sub-cortical regions such as the amygdala, which are important in both experiencing and remembering sensations of certain experiences [21]. Therefore, artificial intelligence needs the capability to have this kind of proprioceptive feeling, which is an internal detection of self-emotional processing and a basis for processing and responding to external emotional information.

### 4.4. Capability to Integrate and Process Multiple Information

Mature empathy is metacognitive: one is aware of empathy—that is, one feels pain, and knows that it is a reaction to the misfortune of others, but not his or her own misfortune [4]. "Traditional view holds that empathy involves 'feeling' the experience of another person's perspective by interpreting his or her emotional expressions (e.g., facial expressions, body language, speech). This is what is called the difference between empathy, sympathy and compassion. Empathy refers to feel the emotions of others through personal experience. Sympathy and compassion do not imply being able to feel the emotions of others based on shared experiences" [3]. This makes the processing capacity of empathy more complex than that of sympathy and compassion and requires the ability to integrate and process multiple types of information. Through the combination of top to bottom (upper and lower) integration as well as internal and external integration of artificial empathy perception, the burden of the central processing system is reduced, and the efficiency of machine emotion perception is improved, thereby it can realize direct, efficient and seamless human–computer interaction.

#### 4.4.1. Empathy for Top to Bottom Integration

Bottom-up emotion generation and regulation processes are intertwined with top-down emotion processing (including emotion adjustment, situational assessment and emotion control). Bottom-up information update and top-down information feedback express emotion. The contribution of the body to emotion should be distinguished along these complementary pathways of brain and body interaction, that is, bottom-up interoceptive signals travel along afferent pathways and top-down autonomic drives travel along efferent pathways to change the internal state of the body [15]. Children's processing of guilt is processed at two different levels, including the experience and performance of guilt, and the ability to report, attribute, and reflect on guilt experiences [22]. This comprehensive ability means that the multi-dimensional interaction of emotions leads to the generation of personal ability in the process of emotional development. At the same time, the emergence of empathy ability is also inseparable from this experience and processing. The brain supports the entire body by predicting or inferring the energy needed to effectively manage future environments, which is based on the necessity to maintain long-term homeostatic balance through interoceptive inputs from internal organs, weighing expectations (beliefs), and responses to previous experiences [15]. That is to say, the brain and the body have the same

position. The body provides experience, and the brain provides cognition, so as to realize emotional processing and empathy under the joint action of experience and cognition.

### 4.4.2. Empathy for Internal and External Integration

The integration ensures that both internal and external emotions can be directly perceived and expressed. Emotional experience is an individual's interpretation and evaluation of the perceived situation, emotional state and expression [18]. If explicit experience and implicit experience are distinguished, emotional experience may occur at different levels of consciousness [18]. Emotion is a life activity that exists as a psychosocial existence and it confronts ongoing problems of both self (internality) and collective (externality) at the same time [3]. Therefore, the internal processing is mainly the processing of the internal feeling and experience of one's own emotions, while the external perception is mainly the perception and expression of the emotions of others, and the two are unified in the overall processing of the body. In conclusion, embodied theory allows scientists to imagine cognitive content in a new way, while directly linking affect and affective knowledge, as it removes the distinction between conception and perception/interoception [5]. Babies have to compare a stranger's face to their internal representation or memory of faces, and fear arises when the face is found to be inconsistent with all other faces in their memory [18]. The external input works with the internal pattern system to reconstruct the emotional experience of the body. Emotional expression and recognition are facilitated when the somatosensory resonance between self and others´ bodies is enhanced [13]. Therefore, resonance is the physical basis for realizing human–machine empathy, and artificial intelligence needs physical resonance to achieve emotional empathy. In the process of human development, autonomous efferent drives maintain homeostasis, not only by generating contextual responses to incoming interoceptive bodily cues, but also by anticipating physiological challenges emanating from extrinsic motivational and emotional cues [15]. Concepts are temporary structures based on entity simulations in working memory, rather than stable structures stored in long-term memory. One possibility is that concepts can be defined as statistical patterns of sensor–motor systems that take different forms in different contexts, and that human conceptual systems have evolved to support specific actions in the environment [21].

### 4.4.3. Empathy for Feedback Processing

Emotional embodiment weakens the role of emotional cognition and computational regulation mechanisms based on computation and representation and increases the role of empathy processing mechanisms based on causal reasoning. At the same time, according to the perception–expression system of human empathy, a dual-loop feedback mechanism of artificial empathy perception–feedback and perception–expression integration is constructed. The key to this is the feedback ability of emotional awareness. How do we recognize emotions of others? One source of information may be the facial feedback signals that arise when we automatically imitate the facial expressions of others [10]. Thus, the feedback is the reverse processing of the previous emotional experience, which can abstract the previous experience and, on this basis, strengthen the previous experience, so as to realize the connection between the concept and experience of abstract emotion. The study found that reducing facial feedback appears to have broad functional effects on emotional processing, including emotional response and emotional perception, as well as a bidirectional effect of facial feedback. When feedback is suppressed, emotional perception accuracy decreases; however, when feedback is enhanced, emotional perception accuracy is improved [10]. The emotional perception accuracy is based on the feedback mechanism from two aspects of physical experience and abstract concept extraction. Situational conceptualization is based on real experience, so when a person experiences an event related to the knowledge, it is likely that conceptualizing knowledge will also activate memory related to the event [19]. For emotional feedback processing, it is the two-way adjustment of image schema formed by embodied simulation experience in the process of emotional perception

and emotional expression. For emotional design of artificial intelligence, feedback processing empathy promotes the accuracy of perception, on the one hand. On the other hand, it improves the consistency of emotional expression, which is the emotional adaptation effect of artificial intelligence in the process of human–computer interaction. "Empathy presents in the experience of using Elefriends, and empathy practices are the key of emotional support users feel, however, empathy can become difficult over time. A person's ability to provide support to others is determined by the degree of distress, and empathy can have an impact on self-care and support practices" [3]. Therefore, feedback-processing empathy will bi-directionally evaluate the previous emotional experience and processing, thereby promoting the continuous synergy of artificial empathy in the process of human-computer interaction. The assessment is carried out by sensation, and is transformed into a preconscious sensation and then into a conscious "feeling of sensation", according to the intensity of emotion. The latter can be described as an intrinsic perception [25]. From an evolutionary point of view, emotion is considered to be evolved to adapt to life, and the basis for its application lies in the moderating effect of feedback processing.

## 5. Conclusions

In Aristotle's view, emotion is the core of a good life. Therefore, realizing human-machine empathy is also a key in realizing human–machine coexistence and integration. However, it is important to note that emotion-related AI, or artificial empathy-based AI, can also be an uncontrollable monster if it is used incorrectly or misunderstood. The scope of data extraction, capture and dissemination by digital technology is blurring the traditional concepts of internality, externality, individuality and collectiveness [3]. If artificial emotion technology is not rationally restricted and is allowed to develop, it may cause devastating damage to human emotional life and lifestyle. Therefore, artificial empathy needs to further analyze human emotional morality and the adaptive scope of artificial emotion. At the same time, it should be pointed out that the occurrence of artificial empathy is a development and co-evolution with human beings. The realization of artificial emotion relies on the simulated "emotional self" generated by a machine's empathy ability, which eliminates the individualization of traditional emotion and the concept of emotional dissociation. Thus, emotion is regarded as a kind of emergence in the interaction relationship, and the emergence of artificial empathy can accelerate the emotional landing (realization) of human–machine integration and artificial intelligence ethics to some extent. Of course, the realization of artificial empathy and the implementation of artificial emotions depend on the improvement of perception ability in the process of artificial intelligence development, as well as the internal simulation of human emotional expression in this technology, so as to achieve consistency with human emotions.

**Author Contributions:** Writing—original draft, Z.C.; Writing—review & editing, Z.C. and J.L. All authors have read and agreed to the published version of the manuscript.

**Funding:** This study was supported by the National Philosophy and Social Science Foundation of China (Grants Nos. 21CZX020 and 20BZX030) and Social Science Foundation of Jiangsu (Grant No. 20ZXC004).

**Institutional Review Board Statement:** Not applicable.

**Informed Consent Statement:** Not applicable.

**Data Availability Statement:** Not applicable.

**Conflicts of Interest:** The authors declare no conflict of interest.

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
