# Peer review of "A Study on Two Conditions for the Realization of Artificial Empathy and Its Cognitive Foundation"

_philosophies, doi:10.3390/philosophies7060135_

Round 1
Reviewer 1 Report
Thank you for allowing me to review this paper. Although I believe that it can make a contribution to the discourse on AI and emotions, I recommend a revise and resubmit. Please refer to the attached PDF of the authors paper with my comments.

Author Response
Reviewer. 1:
Question1: Additionally, there seems to be a problem with the implication here from the statement about the two conditions necessary for the realization of AI empathy to this statement here. AI only needs to simulate empathy for AI empathy. Reflection is not necessary. If AI is simply able to identify emotions and give the appropriate responses which make them seem like they are being empathetic is all that would be (and perhaps could be) necessary.
Perhaps the authors are attempting to make an argument for the limits of AI by attempting to argue for conditions of AI empathy which AI cannot meet? If so, this seems to be problematic for the reasons noted above. In other words, there's a question as to whether the authors are engaging the same question as those who believe AI empathy is possible.
The authors might consider being more explicit about their aim here if it is to suggest the limitations of AI, which may introduce a topic of concern that is new to the discourse on AI empathy, one that is especially philosophical.
The authors might also consider splitting their paper into two kinds of paper. One which addresses the contributions in their paper that is directly and explicitly related to AI empathy research by those who believe in the possibility (perhaps through simulation and not realization of non-AI empathy conditions), and one that focuses more on the philosophical question of the possibility of AI empathy when understood not as simulation but as the realization of the conditions of non-AI empathy.
Answer1. We are grateful for the suggestions. This paper involves two aspects of artificial empathy, namely, the two conditions for the realization of artificial empathy, and the cognitive basis for the realization of artificial empathy which needs to satisfy three cognitive processing abilities (the overall processing ability of external emotions, the overall processing ability of proprioceptive emotions, and the integrated processing ability of multiple information). Therefore, we do not try to divide this study into two papers to write.
Question2: “I would replace "on" with "of," and the authors might want to see if similar edits ought to be made throughout their paper.”
Answer 2: Thanks to your suggestion, we have revised it in the paper.
Question3: The phrasing of these elements do not make them very clear in terms of what the authors can mean with each of the elements listed. For example, by "consistency on external emotion processing and emotional expression," do they mean something like consistency between the identification of external emotions and external, emotional expressions? Or do they mean something consistency between the identification of external emotion processing and "internal" emotional expression. I am assuming the first, but notice the difference in precision between the authors phrasing and the one given here. I advise that the authors go through their entire paper to ensure that each of their sentences are as reasonablly precise as they can be.
Answer3: Thanks to your suggestion. We have changed external emotion processing to identification of external emotions.
Question4: I understand that the authors are attempting to motivate their paper here, but it's not clear why these are "shackles." As they note above, the problem has been a difficulty with the complexity of studying emotions in group interactions.
Answer4. As mentioned in our paper, research on empathy is actually an important response to the individualism and cognitivism in the western tradition. Emotion is not an individual and cognitive-based processing ability, and empathy is more manifested in emergence. In line 89 of the original text, we put forward "emotion is the emergence of social interaction". And, we have marked it in green as stated in Section 2.1.
Question5: It seems as if this is paragraph is also an "introduction." I advise the authors to go through each of their sections to ensure that each section is more focused on the aim of that section. It seems like the authors believe that they need to motivate even their subsections independently. However, a well written (focused) paper should discuss the motivations for the paper in the introduction, and each subsequent section should aim at following through with fulfilling the specific aim of the paper. In doing so, the paper, with all its sections, would constitute an integrated whole. In contrast, this paper seems as if it is constantly "starting over" again and again, with each section.
Answer 5: We have deleted the two sentences in this part to avoid the repetition of the expression, and we have fully revised the whole paper to make the logic and expressions of the paper more proper and complete.
Question6: The authors seem to believe that they made a case for the "individual-centered" state of "current affective computing," but I don't see anything in the previous passages that they have done so. Given this, the reasoning employed by the authors in this passage seems a bit problematic. Perhaps the authors believe that the emphasis on rationality in AI research suggests the use of a "self-centered" model, but this is not necessarily the case and does not warrant the implication from rationality-centered to self-centered. I suggest that the authors go through their entire paper to tighten up the reasoning herein.
Answer 6: We have made further revisions to the paper, and at the same time modified the content in accordance with the title.
Question7: I don't know what this sentence is stating. One reason is grammar and such, but another is that the authors seem to presuppose that the readers have shared terminology. For example, it's unclear to me what the authors mean by "traditional category."
Answer 7. We are grateful for the suggestion. We did not list them in detail in the paper, so we made further revisions in the revised version. The traditional classification refers to dividing emotions into basic emotions and complex emotions (social emotions), but this study further distinguishes emotions and believes that emotions can be divided into three categories, namely basic categories, secondary categories and abstract categories.

Reviewer 2 Report
The paper seems to be a literary review (focusing especially one source) on the role of emotions in a digital context, with several proposals on the possible artificial representations of human emotions and their relations with the present AI. The main part of the paper is a (relativelly obscure) overview on the identification and different kind of classification of the emotions.
Unfortunatelly, reading of the text can not help too much in a better understanding of this interesting topic. I have a feeling, that it is not too hard to find a clearer and better introduction of the field. But it would be okay if this concrete interpretation and description would have any specific conclusion - but it is not included in the text.
However, the selection of the problems (expressed in the structure and the subtitles of the paper) seems to be valuable. Perhaps careful analyses of these problems would be really important for the realization of any kind of artificial emotions. From this point of view the paper can be considered as a (not well argumented) proposal for the future researches in the field.
In this respect I would have a suggestion: perhaps specific ethological researches on the animal emotions (e.g. companion dogs) would be interesting and perhaps more handy to apply in the human computer interactions. Sometimes this research is called "Ethorobotics" and it has several preliminary conclusions. Its philosophical perspective appears in a very interesting and characteristic form in the works (and their development) of Donna Haraway. Her ideas are significant, exciting, and excellent statements.
Author Response
The paper seems to be a literary review (focusing especially one source) on the role of emotions in a digital context, with several proposals on the possible artificial representations of human emotions and their relations with the present AI. The main part of the paper is a (relativelly obscure) overview on the identification and different kind of classification of the emotions.
Unfortunatelly, reading of the text can not help too much in a better understanding of this interesting topic. I have a feeling, that it is not too hard to find a clearer and better introduction of the field. But it would be okay if this concrete interpretation and description would have any specific conclusion - but it is not included in the text.
However, the selection of the problems (expressed in the structure and the subtitles of the paper) seems to be valuable. Perhaps careful analyses of these problems would be really important for the realization of any kind of artificial emotions. From this point of view the paper can be considered as a (not well argumented) proposal for the future researches in the field.
In this respect I would have a suggestion: perhaps specific ethological researches on the animal emotions (e.g. companion dogs) would be interesting and perhaps more handy to apply in the human computer interactions. Sometimes this research is called "Ethorobotics" and it has several preliminary conclusions. Its philosophical perspective appears in a very interesting and characteristic form in the works (and their development) of Donna Haraway. Her ideas are significant, exciting, and excellent statements.
Answer 1: Thank you very much for the suggestions, which are very detailed, and we have learned a lot from them. It is presented that our argument is not too clear, we have revised and further argued in the paper, see red font for details in the revised version of the paper.
Answer 2: For the animal emotions mentioned in the suggestion, we greatly appreciate it. We do not think the analogy about the emotional relationship between animals and humans is appropriate, and we have deleted the analogy from the text. Therefore, we do not study animal emotions in this paper, because animals and machines are not comparable. Donna Haraway mentioned in the paper focuses more on social criticism (the relationship between machines and people), and less on exploration of the mind of the machine, which has a certain distance from the research in this paper, so we are not prepared to introduce the analysis of "Ethorobotics" into this paper. Our future research will study from this perspective. Thank you very much for your suggestion.

Reviewer 3 Report
- This article deals with the topic of ‘artificial empathy.’ Since studies of AI are mainly focused on intelligence, the topic of this article is worth pursuing. As a philosopher who is interested in the nature of emotion, feeling, and consciousness, the choice of the topic attracted my attention. However, I found several holes, both in the structure and in the arguments. On the one hand, for the article to be coherent, the notion of empathy should be properly distinguished. On the other hand, insofar as the article attempts to investigate the phenomena of emotion, it must be based on the scientific theory of consciousness.
- First, the concept of empathy must be clarified. It is widely accepted that there are two kinds of empathy: cognitive empathy and emotional empathy. Roughly put, when one recognizes or infers another’s emotion, it is called "cognitive empathy. Cognitive empathy is about "knowing" how others feel. If one actually feels what the other feels, it is called emotional empathy. Emotional empathy is about "feeling" how others feel. In the case of cognitive empathy, one does not need to actually share another’s feelings. Emotional empathy, however, requires one to share another's emotional experience. It is easy to see that cognitive empathy is not necessarily accompanied by emotional empathy. Given this distinction, artificial empathy should be distinguished. There can be artificial cognitive empathy and artificial emotional empathy. As two empathies are different in their nature, these artificial empathies must not be confused with each other. An article's direction and argumentation will depend on which empathy is its main subject.
- The problem is that I could not find any clue about that question. The article merely says that it is about the realization of artificial empathy without clarifying what kind of empathy it attempts to address. When the article says that it is crucial for AI to identify human emotions (line 12), it seems that it wants to address the realization of artificial cognitive empathy. When it says it is important to "prove AI has the ability to reflect on emotional consciousness(line 12), the article seems to include not only cognitive empathy but also emotional empathy. In fact, I do not fully understand what "the ability to reflect on emotional consciousness" exactly means. If it means AI’s ability to access and reflect on its own emotional experiences, the question of whether AI can have emotional experiences at all must be addressed first. Unfortunately, as far as I can tell, the article gives no further explanation on the question of AI’s experience or consciousness.
- Whether the article's topic is cognitive empathy or emotional empathy, it is crucial. If the article wants to focus on artificial cognitive empathy, it does not have to deal with "the ability to reflect on emotional consciousness", because cognitive empathy has nothing to do with emotional consciousness or reflection upon it. To cognitively empathize with someone else’s feelings, AI does not have to be able to have an emotional experience of its own or reflect on that experience. All AI needs is the ability to recognize and infer what kinds of emotions humans are feeling and to present appropriate behavioral responses. Indeed, this procedure can be done totally without reflection on emotional consciousness. In other words, even emotional zombies can perfectly do emotional labor if they are properly programmed and trained by emotion data. The authors seem to think that to present emotion that is consistent with human emotion, AI must have the ability to reflect on emotional consciousness. I do not see why it is so. Perception and expression of emotion can be fully captured in functional, informational, or cognitive terms. That is, both proper understanding and consistent presentation of emotion can be covered by implementing artificial cognitive empathy. No consciousness, no reflection is required. If so, Section 4 is redundant or misguided at best, as it is based on the assumption that in order for AI to present emotions consistent with human emotions, it must be able to reflect on its own emotional experiences.
- Even if the article attempts to address the issue of artificial emotional empathy, there remains a huge issue. The realization of emotional empathy requires AI to have the same kind of emotion as a human. This requirement follows directly from the very definition of emotional empathy. The problem is that emotion is one of the clear examples of conscious experience. Although there might be some sense in talking about unconscious emotions, in the context of empathy and human-AI interaction, emotions are taken as conscious mental states that we humans can be easily aware of. Here, artificial emotions are not merely functional equivalents of human emotions. They are the same kinds of emotions that humans consciously feel. If the realization of artificial emotional empathy is at issue, the possibility of artificial emotion should be addressed first, and to address such a possibility, the question of whether AI can have the same kind of consciousness that we have must be answered. It appears that in Section 4, the authors are trying to deal with this issue. However, without the solid foundation of the science of consciousness, all the talk of "the processing of ontological emotions and external emotions, the integration of internal and external emotions, and the generation of proprioceptive emotions" is too weak to support the authors’ point. It might be the case that when such conditions are met, AI can have emotional consciousness as we do and reflect on that consciousness. But why should that be the case? How can that be the case? The generation of emotional consciousness must be explained, not presupposed. To explain why and how such conditions lead to the generation of emotional experiences and reflections, one needs the theoretically plausible and empirically supported science of consciousness. Except for rare mentions and references, I could not find any theoretical or empirical background for the authors’ argument.
Author Response
- This article deals with the topic of ‘artificial empathy.’ Since studies of AI are mainly focused on intelligence, the topic of this article is worth pursuing. As a philosopher who is interested in the nature of emotion, feeling, and consciousness, the choice of the topic attracted my attention. However, I found several holes, both in the structure and in the arguments. On the one hand, for the article to be coherent, the notion of empathy should be properly distinguished. On the other hand, insofar as the article attempts to investigate the phenomena of emotion, it must be based on the scientific theory of consciousness.
- First, the concept of empathy must be clarified. It is widely accepted that there are two kinds of empathy: cognitive empathy and emotional empathy. Roughly put, when one recognizes or infers another’s emotion, it is called "cognitive empathy. Cognitive empathy is about "knowing" how others feel. If one actually feels what the other feels, it is called emotional empathy. Emotional empathy is about "feeling" how others feel. In the case of cognitive empathy, one does not need to actually share another’s feelings. Emotional empathy, however, requires one to share another's emotional experience. It is easy to see that cognitive empathy is not necessarily accompanied by emotional empathy. Given this distinction, artificial empathy should be distinguished. There can be artificial cognitive empathy and artificial emotional empathy. As two empathies are different in their nature, these artificial empathies must not be confused with each other. An article's direction and argumentation will depend on which empathy is its main subject.
Answer 1: Many thanks to your suggestions. We have added the notion of empathy and its three categories to the Introduction part. And we divide artificial empathy into three categories: (1) Action empathy, an automatic non-cognitive process, that manifests as the convergence of machines to human emotions in expressions, voices, and gestures; (2) Emotional empathy, which is human emotional state-oriented emotional responses machine uses; â‘¢ Cognitive empathy, the ability of machines to recognize human emotional states, and to transpositionally think and reason.
2.- The problem is that I could not find any clue about that question. The article merely says that it is about the realization of artificial empathy without clarifying what kind of empathy it attempts to address. When the article says that it is crucial for AI to identify human emotions (line 12), it seems that it wants to address the realization of artificial cognitive empathy. When it says it is important to "prove AI has the ability to reflect on emotional consciousness (line 12), the article seems to include not only cognitive empathy but also emotional empathy. In fact, I do not fully understand what "the ability to reflect on emotional consciousness" exactly means. If it means AI’s ability to access and reflect on its own emotional experiences, the question of whether AI can have emotional experiences at all must be addressed first. Unfortunately, as far as I can tell, the article gives no further explanation on the question of AI’s experience or consciousness.
Answer 2: We are very grateful for your suggestion. For "the ability to reflect on emotional consciousness", we believe that the ability to reflect on emotions is the advanced processing ability of machines for emotional understanding and expression. We will mark it in the paper. For the question of whether AI has emotions, we just presuppose AI's emotional experiences, the purpose of this paper is to demonstrate the possibility of AI having emotional experiences, so it cannot be said in the paper that it really has.
3- Whether the article's topic is cognitive empathy or emotional empathy, it is crucial. If the article wants to focus on artificial cognitive empathy, it does not have to deal with "the ability to reflect on emotional consciousness", because cognitive empathy has nothing to do with emotional consciousness or reflection upon it. To cognitively empathize with someone else’s feelings, AI does not have to be able to have an emotional experience of its own or reflect on that experience. All AI needs is the ability to recognize and infer what kinds of emotions humans are feeling and to present appropriate behavioral responses. Indeed, this procedure can be done totally without reflection on emotional consciousness. In other words, even emotional zombies can perfectly do emotional labor if they are properly programmed and trained by emotion data. The authors seem to think that to present emotion that is consistent with human emotion, AI must have the ability to reflect on emotional consciousness. I do not see why it is so. Perception and expression of emotion can be fully captured in functional, informational, or cognitive terms. That is, both proper understanding and consistent presentation of emotion can be covered by implementing artificial cognitive empathy. No consciousness, no reflection is required. If so, Section 4 is redundant or misguided at best, as it is based on the assumption that in order for AI to present emotions consistent with human emotions, it must be able to reflect on its own emotional experiences.
Answer 3: This paper mainly demonstrates the two conditions and cognitive basis for the realization of artificial empathy from the perspective of philosophy and psychology, but not from the perspective of computer science to study artificial empathy itself. In addition, we talk about artificial empathy as a whole, so this paper does not distinguish the various levels of artificial empathy. Thanks to your suggestion, we will explore this in our future research.
4.- Even if the article attempts to address the issue of artificial emotional empathy, there remains a huge issue. The realization of emotional empathy requires AI to have the same kind of emotion as a human. This requirement follows directly from the very definition of emotional empathy. The problem is that emotion is one of the clear examples of conscious experience. Although there might be some sense in talking about unconscious emotions, in the context of empathy and human-AI interaction, emotions are taken as conscious mental states that we humans can be easily aware of. Here, artificial emotions are not merely functional equivalents of human emotions. They are the same kinds of emotions that humans consciously feel. If the realization of artificial emotional empathy is at issue, the possibility of artificial emotion should be addressed first, and to address such a possibility, the question of whether AI can have the same kind of consciousness that we have must be answered. It appears that in Section 4, the authors are trying to deal with this issue. However, without the solid foundation of the science of consciousness, all the talk of "the processing of ontological emotions and external emotions, the integration of internal and external emotions, and the generation of proprioceptive emotions" is too weak to support the authors’ point. It might be the case that when such conditions are met, AI can have emotional consciousness as we do and reflect on that consciousness. But why should that be the case? How can that be the case? The generation of emotional consciousness must be explained, not presupposed. To explain why and how such conditions lead to the generation of emotional experiences and reflections, one needs the theoretically plausible and empirically supported science of consciousness. Except for rare mentions and references, I could not find any theoretical or empirical background for the authors’ argument.
Answer 4: Artificial emotion is not an independent emotion, but an emotional state that emerges in the process of interacting with machines. For details, please refer to “emotion is the emergence of social interaction” in section 2.1 (Line 89).

Round 2
Reviewer 2 Report
Thanks for the clarifications. Now I think the paper is better and acceptable.